# Neuron Circuit Failure and Pattern Learning in Electronic Spiking Neural Networks

Sumedha Gandharava, Robert C. Ivans, Benjamin R. Etcheverry and Kurtis D. Cantley *

Department of Electrical and Computer Engineering, Boise State University, Boise, ID 83725, USA;
sumedha248@gmail.com (S.G.); robert.ivans@inl.gov (R.C.I.); benetcheverry@u.boisestate.edu (B.R.E.)
* Correspondence: kurtiscantley@boisestate.edu

**Abstract:** Biological neural networks demonstrate remarkable resilience and the ability to compensate for neuron losses over time. Thus, the effects of neural/synaptic losses in the brain go mostly unnoticed until the loss becomes profound. This study analyses the capacity of electronic spiking networks to compensate for the sudden, random neuron failure ("death") due to reliability degradation or other external factors such as exposure to ionizing radiation. Electronic spiking neural networks with memristive synapses are designed to learn spatio-temporal patterns representing 25 or 100-pixel characters. The change in the pattern learning ability of the neural networks is observed as the afferents (input layer neurons) in the network fail/die during network training. Spike-timing-dependent plasticity (STDP) learning behavior is implemented using shaped action potentials with a realistic, non-linear memristor model. This work focuses on three cases: (1) when only neurons participating in the pattern are affected, (2) when non-participating neurons (those that never present spatio-temporal patterns) are disabled, and (3) when random/non-selective neuron death occurs in the network (the most realistic scenario). Case 3 is further analyzed to compare what happens when neuron death occurs over time versus when multiple afferents fail simultaneously. Simulation results emphasize the importance of non-participating neurons during the learning process, concluding that non-participating afferents contribute to improving the learning ability and stability of the neural network. Instantaneous neuron death proves to be more detrimental for the network compared to when afferents fail over time. To a surprising degree, the electronic spiking neural networks can sometimes retain their pattern recognition capability even in the case of significant neuron death.

**Keywords:** spiking neural network; spike timing-dependent plasticity; memristor; spatio-temporal pattern recognition

## 1. Introduction

Neuron death occurs in biological neural networks (the brain) due to various reasons such as aging, natural death during migration and differentiation, head injuries, spinal cord injuries, or neurodegenerative diseases. Cognitive functionality of the human brain gradually declines with age leading to memory loss, learning slowdown, motor incoordination, and attention impairment [1,2]. Neurodegenerative diseases also cause a considerable decline in neuron numbers. Parkinson's and Huntington's diseases lead to neuron death in the basal ganglia region of the brain, and Alzheimer's affects the neurons in the neocortex and hippocampus [1,3,4]. It generally takes about 60 years before people notice any measurable memory loss or become susceptible to develop neurodegenerative diseases [3]. Thus, the human brain demonstrates a remarkable ability to compensate for neuron losses over time, forestalling any noticeable effect until the losses become profound [2,5]. According to one study from 1998, about 11 million people in the US experienced a stroke, of whom only approximately 0.77 million (7%) were symptomatic [6]. A vast majority of strokes are 'silent', although they can kill large numbers of cells rapidly [6]. While some work has shown recurrent networks to be more robust to neuron death, presently, the network-level

effects of neuron failure (or death) in feed-forward networks are largely not addressed in the scientific literature [7,8]. This study contributes to filling a gap in neuromorphic computing research by analyzing the resilience of spiking neural networks (SNNs) in which neuron death occurs.

Pioneering research has demonstrated artificial neural networks for a broad range of applications. Future systems are expected to use pulses or spikes instead of analog signals to communicate and transfer information to achieve higher levels of cognition. Customized hardware implementations will make these spiking neural networks (SNNs) not only highly efficient, but also robust and fault-tolerant. SNNs are expected to find applications in harsh, radiation-filled environments such as space or at nuclear and military installations. Presently, shielding and hardening are common practices to protect devices and circuits from radiation, but these techniques are unable to block all particles from interacting with underlying electronics [9,10]. Radiation in such cases can lead to neuron circuit failure through various mechanisms including CMOS threshold voltage shift, oxide breakdown, gate rupture, and displacement damage [11,12].

In this article, a memristor-based SNN is trained to recognize a spatio-temporal pattern, and changes in the recognition accuracy of the networks due to the death of neurons are analyzed. In the SNN, synapses are realized using a memristor behavioral model. Unlike other non-volatile memories, memristors do not need to be refreshed to maintain their state, and this decreases the power consumption of the system. Long-term degradation of memristive devices most often results in off-state resistance changes and not complete loss of functionality. To a certain degree, many memristors are made from materials that are relatively resistant to radiation effects [13–16]. The possibility of constant training or occasional refreshing of synaptic weights along with a typically large number of synapses per neuron therefore makes memristive synapses less of a concern from a system reliability standpoint. Spike-timing-dependent plasticity (STDP) is a biological learning process that alters the synaptic weight depending on pre- and post-synaptic neuron firing time. The STDP rule is implemented in the presented network using biphasic shaped spikes from the pre- and post-synaptic neurons that enforce the change in the conductivity/synaptic weight of the memristor depending on their activity. Although the network uses a single layer, the results can provide insight into the operation and response of other topologies such as spiking convolutional deep neural networks [17].

Section 2 in the article describes the SNN design, memristor behavioral models, and the design of the leaky integrate-and-fire (LIF) post-synaptic neuron circuits. Section 3 outlines the experimental setup and the results obtained from simulations for the three cases outlined previously. Section 4 concludes by discussing the significance of the results and the future implications and applications of the work.

## 2. Simulation Methods

This configuration of the network used for simulations is detailed in Section 2.1. Section 2.2 describes the modified memristor model used to implement the synaptic behavior (learning rule) in the neural network. Section 2.3 discusses the post-synaptic neuron designed based on the leaky integrate-and-fire (LIF) circuit behavior. The simulations are performed in the Cadence Spectre and the memristor and LIF behavioral model is designed in Verilog-A.

Interventionary studies involving animals or humans, and other studies that require ethical approval, must list the authority that provided approval and the corresponding ethical approval code.

### 2.1. Neural Network Topology

The network used in this study, shown in Figure 1, consists of multiple pre-synaptic neurons (afferents) and one post-synaptic (output) neuron, all connected through memristive synapses. Synapses act as the memory element and create a connection between the initial and the final layer of the network. This network topology is a single-layer feed-forward

network with either 25 or 100 pre-synaptic afferents ($N_1$ to $N_{25 \text{ or } 100}$), each connected to a single post-synaptic afferent via single memristor ($M_1$ to $M_{25 \text{ or } 100}$ in Figure 1). The network uses biphasic shaped pulses to achieve pair-based spike time-dependent plasticity (STDP) for pattern learning, discussed further in Section 3. More information about the network architecture can be found in refs. [18–20]. Neuron death in the network is imitated by disabling pre-synaptic neurons randomly during the simulation (STDP learning happens constantly). Failure of the output neuron circuit is not considered in any case, since loss of output would eliminate any ability to perform analysis.

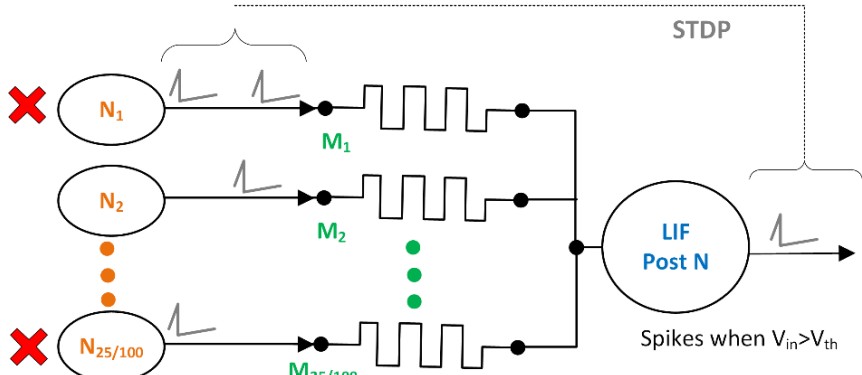

**Figure 1.** Memristor-based electronic SNN architecture used in this study for spatio-temporal pattern recognition. Either 25 or 100 pre-synaptic neurons (afferents) are connected to one post-synaptic leaky integrate-and-fire (LIF) neuron via single memristors. The network uses biphasic shaped pulses to achieve pair-based STDP for pattern learning. Random neuron death is simulated by disconnecting pre-synaptic neurons after 30 s of partial learning.

*2.2. Memristor Modeling*

Memristors are two-terminal devices that can hold their present resistance state until sufficient external bias is applied to change their conductivity. This study uses a $TiO_2$ based voltage controlled ionic drift memristor model, shown in Figure 2a. The $TiO_2$ based non-linear ionic drift memristor model was proposed by Strukov et al. in 2008 [21]. Although the model has lower accuracy, it has been widely used in simulations and comparison studies by Chua et al. for designing networks with memristive bridge synapses and others [22–27]. The model is not purely mathematical, has explicit I-V relationship, includes non-linearity, and has normalized state variable $\left(\frac{w}{D}\right)$.

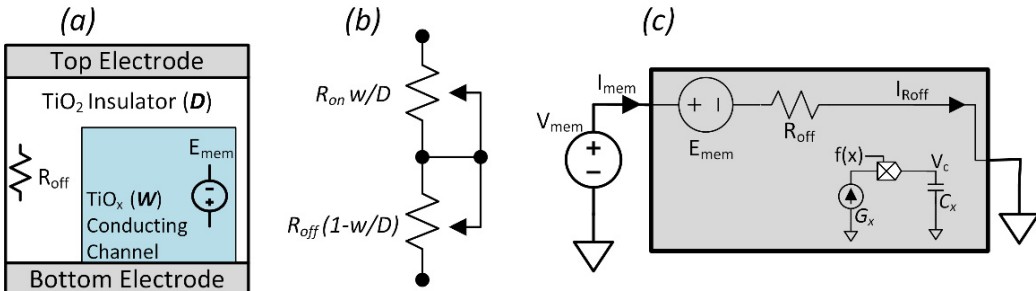

**Figure 2.** (**a**) Diagram of a $TiO_2$ resistive memory (memristor) device. The $TiO_2$ insulator layer of thickness $D$ ($R_{off}$) is between two conductive electrodes. The presence of oxygen vacancies resulted in the formation of the varying conducting channel of width 'w' as $E_{mem}$ in the model. (**b**) Circuit representation of two variable resistors, $R_{on}$ (less resistive region of width w) and $R_{off}$ (high resistive region of width, D−w). (**c**) Synaptic memristive circuit implementation with dependent source $E_{mem}$ and resistance $R_{off}$ and auxiliary circuit with $I_{mem}$ dependent current source $G_x$ and 1 F capacitor $C_x$. The voltage across $C_x$ controls $E_{mem}$.

The model effectively treats the instantaneous total resistance of a memristive device $R_{mem}$ as two variable resistors connected in series, as represented in Figure 2b. One of these resistors represents a conductive region of thickness w inside a device with physical thickness D. The other resistor corresponds to a less conductive region of thickness D−w. When w is almost equal to the device thickness, D, the device is in its lowest resistance state with resistance value $R_{mem}$ equal to $R_{on}$. The device is in a high resistance state with $R_{mem}$ equal to $R_{off}$ when w is much lesser than total device thickness D. The memristor model is accompanied by a window function (f(x)) in Figure 2c that is used to add device-specific non-linearity to the model and also to force the physical boundary of the device [28]. More about the memristor model is detailed in ref. [29].

*2.3. Post-Synaptic Neuron Design*

The post-synaptic neuron used in the biphasic spiking neural network in Figure 1 implements leaky integrate-and-fire (LIF) behavior. The LIF circuit is often realized using operational amplifiers and MOSFETS [30–32]. This work uses an equivalent behavioral model designed in Verilog-A to capture critical aspects of circuit operation (Figure 3a). The LIF circuit fires a bi-directional biphasic spike (toward the dendritic and axonic synapses) when a certain threshold is reached, as in Figure 3b. The schematic depiction of the LIF circuit is presented in Figure 3a. The input of the circuit $V_{PostIn}$ is the node connected to the output of all the memristors in the fully connected network presented in Figure 1. $V_{PostIn}$ depends on the conductivity of the memristors in the network and the spike timing of the pre-synaptic afferent neurons.

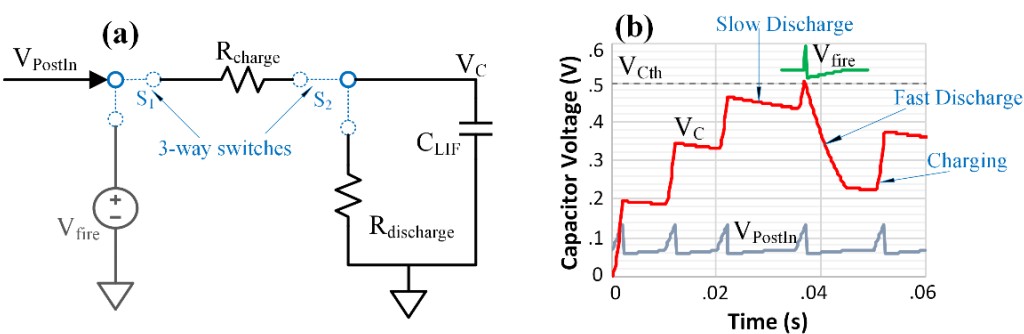

**Figure 3.** (**a**) Leaky integrate-and-fire (LIF) post-synaptic neuron circuit schematic. The circuit is implemented in Verilog-A. Voltage source $V_{fire}$ produces the desired shape of the post-synaptic biphasic spike. $C_{LIF}$, $R_{charge}$, and $R_{discharge}$ mimic membrane leakage in the biological synapse. (**b**) As the circuit sees the input spikes overtime ($V_{PostIn}$), there is an increase in the voltage across capacitor $C_{LIF}$. $V_{fire}$ produces the output spike as the $V_C$ reaches threshold voltage $V_{Cth}$ (= 0.5 V in this case).

**3. Results and Analysis**

The following section presents transient simulations of the neural network and subsequent analysis of changes in its learning capability and pattern recognition accuracy in the event of neuron death. Section 3.1 discusses the spatio-temporal pattern learning approach using correlated and uncorrelated spiking. Section 3.2 provides the general details of neuron death in the simulations for which the results are presented in Section 3.3. Specifically, changes in the pattern recognition ability of the network are tracked when a certain percentage of afferent neurons fail during learning.

*3.1. No Neuron Death (Control Case)*

Before simulating situations in which neuron death occurs, typical network operation is described in this section. Pre-synaptic afferent neurons ($N_1$ to $N_{25 \text{ or } 100}$) in the following simulations fire 10 ms biphasic spikes at an average rate of 5 Hz for the 100 s transient simulation time. Figure 4a shows the time at which spikes arrive for a 25-pixel pattern 'B'.

Afferents that are part of the pattern such as $N_{12}$ and $N_{13}$ fire mutually correlated spikes at regular 200 ms intervals. Although the firing interval is regular for these participating neurons, the exact position of the spike within the 200 ms window is chosen randomly. Non-participating afferents fire uncorrelated (random) spikes with Poisson distributed inter-spike intervals (ISIs), as shown by $N_{14}$ and $N_{15}$ in Figure 4a [33,34]. Lighter color pixels in Figure 4b (at 60 s) are participating neurons and darker ones are non-participating neurons firing uncorrelated spikes. Figure 4b shows the synaptic weight evolution of all the memristors ($M_1$ to $M_{25}$) as the network tries to learn the 25-pixel letter 'B'. Initially, all the weights are set in a highly conductive state. After 30 s of the training, the network was able to depress most of the uncorrelated neurons by decreasing the conductivity of their corresponding memristors, and the desired pattern is very recognizable. At 60 s, the network is in a stable state with post-synaptic neuron firing at a constant rate. Figure 4c shows the synaptic weight distribution of the memristors and corresponding decrease in weight of uncorrelated synapses during learning. More about the network behavior in absence of neuron death can be found in refs. [18,19].

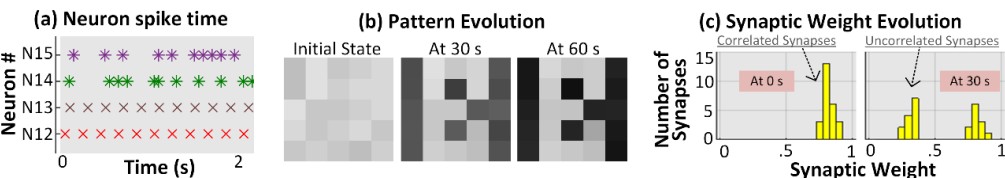

**Figure 4.** (**a**) Scatter plot of spike timing. $N_{12}$ and $N_{13}$ fire mutually correlated spikes at regular 200 ms intervals (participating afferents), while $N_{14}$ and $N_{15}$ (non-participating afferents) fire with random, Poisson distributed inter-spike intervals. (**b**) The synaptic weight evolution of all the memristors ($M_1$ to $M_{25}$) as the network is learning the 'B' pattern. (**c**) Histograms of the synaptic weight distributions in weight bins that are 0.05 wide. After 30 s, uncorrelated neurons are separated and moved to a lower weight.

The neural network used for the remaining parts of this study has 100 pre-synaptic neurons with 60 participating (firing mutually correlated spikes) and 40 non-participating (firing Poisson distributed uncorrelated spikes) afferents. Figure 5 shows the spiking characteristics of the 40 uncorrelated/non-participating pre-synaptic afferents individually and collectively over time. Figure 5a represents the firing times of the 40 uncorrelated afferents for the first 20 s of the simulation. The random ISI distribution is notable. Figure 5b shows the random distribution of firing frequency for each of the 40 uncorrelated afferents with a mean of 5 Hz. Figure 5c captures the population firing rate of all 40 non-participating neurons over the full 100 s stimulation. In this case, frequency is measured in 1 s increments (bin size).

Figure 6 presents the firing frequency of the network over 100 s of simulation. In this case, the population firing rate is measured over reduced 100 ms bins to observe a finer distribution. The 40 non-participating afferents in Figure 6a show random frequency distribution as they are firing Poisson distributed noise. On the other hand, the frequency distribution of 60 participating afferents in Figure 6b is not random, as they are firing mutually correlated spikes at 5 Hz, although at different times relative to the start of each pattern presentation window. Figure 6c shows the population frequency distribution of the whole network over 100 s of the simulation in 100 ms time bins. These data demonstrate that there are no instances of very high afferent population firing rate that would cause overexcitation of the output neuron. It can be noted that, on average, all afferents (participating and non-participating) are firing at the rate of 5 Hz.

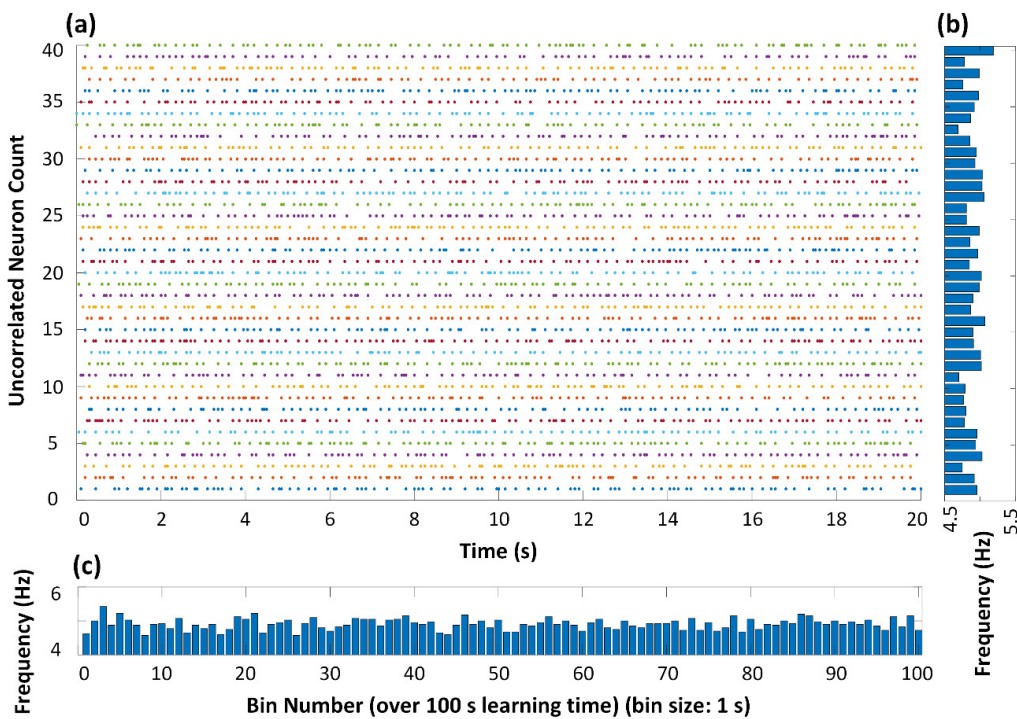

**Figure 5.** A scatter plot of spiking shows the random firing behavior of the 40 non-participating afferents. (**a**) A 20 s snapshot of firing times of 40 non-participating afferents. (**b**) Random distribution of firing frequency for each of the 40 uncorrelated afferents (mean 5 Hz). (**c**) Population firing rate of all 40 non-participating neurons over the full 100 s stimulation with frequency measured in 1 s increments.

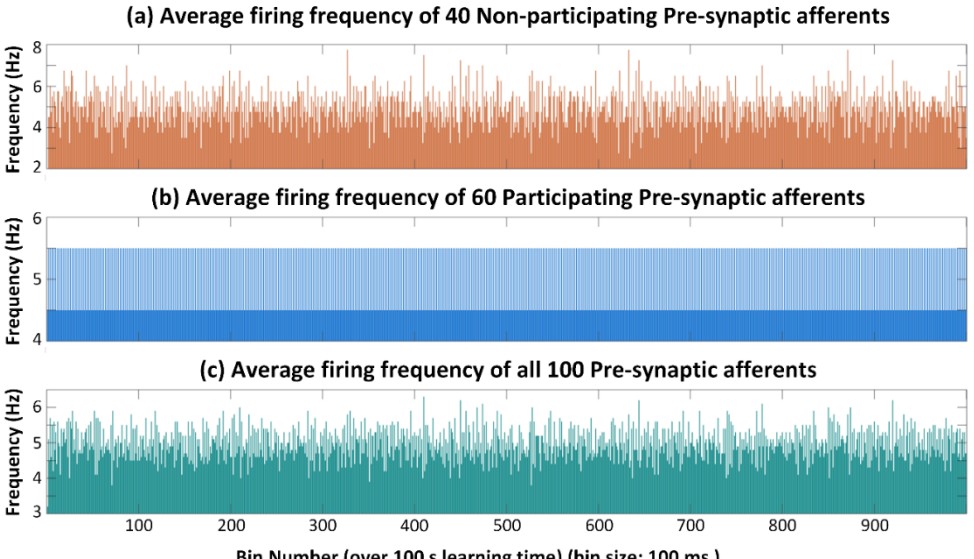

**Figure 6.** Population firing frequency using 100 ms time bins. (**a**) Frequency distribution for the 40 non-participating afferents with Poisson distributed ISIs. (**b**) The frequency distribution of 60 participating afferents is not random, since they are firing mutually correlated spikes. (**c**) Frequency distribution of all afferent neurons over the entire 100 s transient simulation.

Figure 7 shows the response of the post-synaptic LIF neuron during the 100 s learning period. As can be seen in Figure 7a, the post-synaptic neuron is firing periodically every 0.2 s (200 ms) after 30 s, except a few misses which result in 0.3 s, 0.5 s, and 0.6 s intervals. Figure 7b shows the frequency response over 100 s of simulation with a bin size of 5 s.

Initially, the post-synaptic LIF neuron was overexcited due to high equivalent synaptic conductance (strong weights), but stabilized over time to approximately 5 Hz as the network suppresses connections of non-participating afferents.

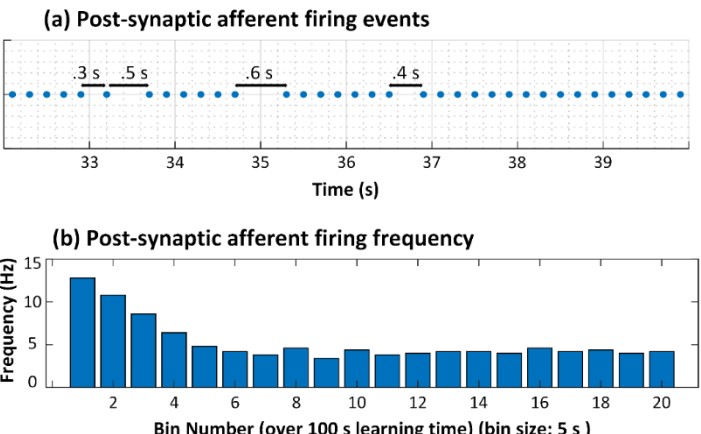

**Figure 7.** (**a**) Post-synaptic neuron firing frequency settles to 0.2 s periods after 30 s, except a few misses. (**b**) Frequency response of the post-synaptic neuron over the 100 s transient simulation (bin size of 5 s) displays how the network stabilizes network over time as non-participating afferent synaptic weights are suppressed.

### 3.2. Neuron Death Setup

The networks analyzed in the following section all use 100 pre-synaptic neurons, out of which 60 neurons are participating/correlated (forming the part of letter B, brighter pixel, firing correlated spikes) and 40 are non-participating/uncorrelated (not the part of letter B, darker pixel, firing uncorrelated spikes). Three neuron death cases are examined in this work. The first case observes the changes in the pattern learning ability of the network when random neurons that fail are all from the set of 60 participating afferent neurons ("participating neuron death"). The second case is that in which random neurons that fail are all from the set of 40 non-participating afferent neurons ("non-participating neuron death"). The third case is most realistic, in which failing neurons are chosen randomly from the set of all 100 neurons, participating and non-participating ("random neuron death"). There are no simulations in which neuron death is initiated prior to 30 s, which gives the network time to at least partially train on the pattern (as depicted in Figure 4b,c). In each case, five sets of randomly chosen afferents are killed to improve the statistical validity of the conclusion.

For each of the three cases in which the described populations of neurons die, two different failure mechanisms are examined which exhibit vastly different timing characteristics. The first is when the given percentage of neurons fail simultaneously at 30 s. Physically, this kind of failure could be caused by an electromagnetic pulse, or from a brief exposure to high fluence ionizing radiation (for example from detonation of a nuclear weapon). On the other hand, a more typical situation would be slow, random failure over time. This could be caused by deployment of the neural network in harsh environments, such as in space or at nuclear waste facilities.

### 3.3. Neuron Death Simulations

One way to examine differences in network response to neuron failure is to examine the inter-spike interval (ISI) of the output (post-synaptic) neuron versus time. Figure 8a shows the ISI produced by the post-synaptic neuron in the case when no neuron death occurs. After 30 s of training, the ISI is essentially always a multiple of 200 ms, at either 0.2 s or 0.4 s. The latter case is indicative of the output neuron failing to respond during a pattern interval, and could be improved by further tuning and training of the network. Figure 8b,c shows the output ISI over time when one neuron fails in the non-participating and participating

groups, respectively. Figure 8d shows a failure of a single neuron occurring from either the participating or non-participating (referred to as combined) groups. After 30 s, the ISI is still approximately 0.2 s or 0.4 s in each of the cases, and almost no change is the pattern learning behavior of the network is observed. However, Figure 8e–g presents the output ISI over time when 50% of randomly selected afferents fail simultaneously at 30 s in the non-participating, participating, and combined groups. After the neuron death at 30 s, no spiking in the post-synaptic neuron is observed and the network was not able to learn the pattern in any of the cases, which is a catastrophic outcome.

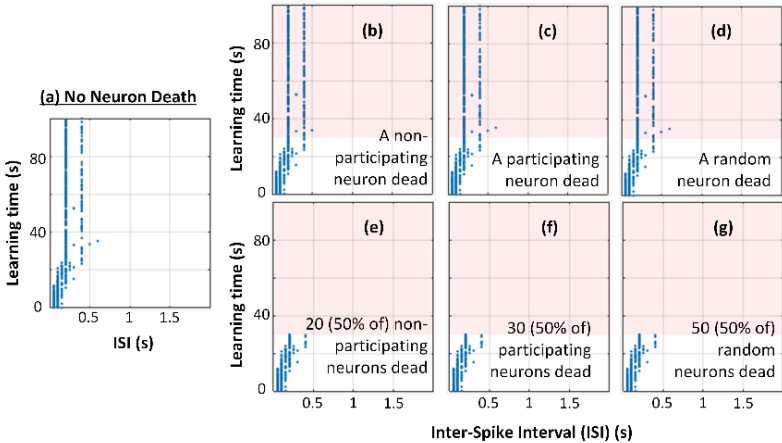

**Figure 8.** Post-synaptic neuron ISI over the learning period for simultaneous neuron death occurring at 30 s, the time following which is indicated by the pink shading. (**a**) ISI is about 0.2 s or 0.4 s in the case when no neuron death occurs. (**b**–**d**) The ISI when one neuron (in groups of non-participating, participating, and combined) fail, in which the network shows no degradation in pattern detection ability. (**e**–**g**) The ISI over time when 50% of randomly selected afferents die (within the participating, non-participating, and combined groups). The network exhibits no post-synaptic neuron activity in any case, thus indicating complete network failure.

The same situation in Figure 8 was simulated five total times with each simulation having a different set of randomly selected afferents disabled (set 1 to set 5) while the input spiking patterns were kept the same. Figure 9a–c shows the ISI over time when 5% afferents in each of the participating, non-participating, and combined groups failed simultaneously at 30 s. Interestingly, failure of either participating and non-participating neurons by themselves did not significantly destabilize the system, as shown in Figure 9a,b. However, network instability is notable in Figure 9c in which neurons from both participating and non-participating groups were eliminated. This is observable in the scatter of the ISIs up to much higher values than 0.2 or 0.4 s. Figure 9d–f shows the ISI over time when 10% of the afferents in the participating, non-participating, and combined groups failed. It is observed in Figure 9e, perhaps unsurprisingly, that non-participating afferent death keeps the system relatively stable. In the case of random neuron death shown in Figure 9f, after 30 s, the post-synaptic neuron is not learning the pattern and the system becomes unstable, as the ISI is again randomly distributed with overall large values. Similarly, Figure 9g–i shows the ISI of a post-synaptic neuron when 25% pre-synaptic neurons in the participating, non-participating, and combined groups failed. Random combined neuron death shows the most instability in this case, as well.

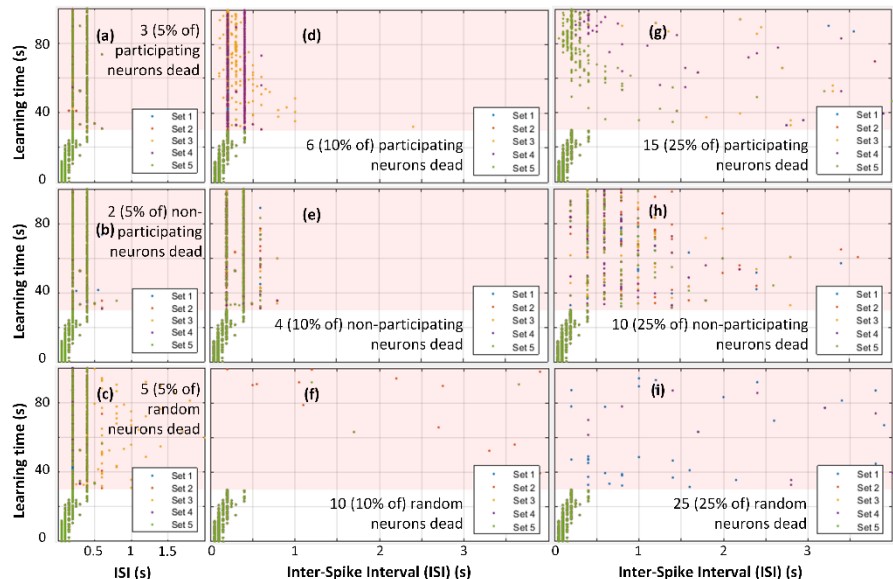

**Figure 9.** Post-synaptic neuron ISI over the 100 s simulation period. Pink shading indicates time after simultaneous neuron death at 30 s. (**a**–**c**) The ISI when 5% of randomly selected neurons fail (grouped as non-participating, participating, and combined). The network was able to recover in both cases (**a**,**b**), but not in (**c**) (random death of both participating and non-participating neurons). (**d**–**f**) The ISI over time when 10% of randomly selected afferents die (within the same three groups). Overall, network stability decreases as afferent death percentage increases, although neuron death in the combined group of participating or non-participating neurons creates the highest degree of network instability. (**g**–**i**) The ISI distribution for each of the three cases when 25% of neurons fail.

Figure 10 shows the normalized average synaptic weight evolution of all the 100 synaptic memristors in the network. Deviation of weights can be observed in the case of random afferent failure Figure 10a) as the 10%, 25%, and 50% failure rates all show deviation from the no-death case. For participating afferent failure Figure 10b), the 25% and 50% failure rates exhibit significant deviation from the no-death case. Finally, non-participating afferent death (Figure 10c) seems to destabilize the system least. In this case, both the 25% and 50% failure rates deviate initially, but the 25% case evolves back towards the no-death situation as the system regains stability (with continued training via presentation of the patterns).

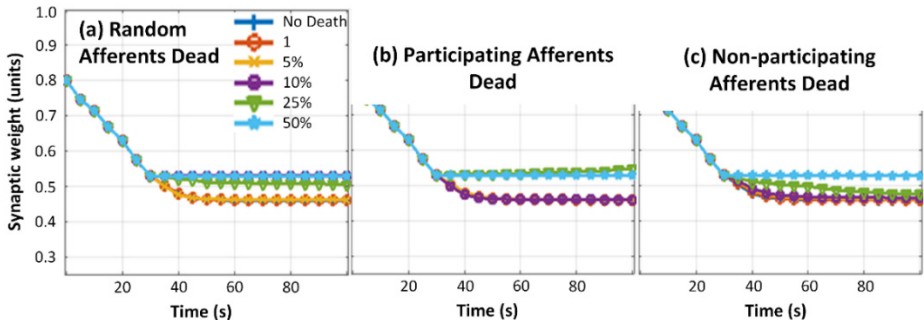

**Figure 10.** Normalized average synaptic weight evolution of all the 100 synaptic memristors in the network. (**a**) In the case of random combined afferent death, 10%, 25%, and 50% death weight evolutions show significant deviation from the no death case. (**b**) In the case of participating afferent death, only the 25% and 50% death weight evolutions show deviation from the no death case. (**c**) Finally, in the case of non-participating afferent death, only 50% death weight evolution shows deviation from the no death case. The 25% death case in fact deviates initially, but then recovers to regain system stability.

The recovery seen in the average synaptic weight in the network with 25% non-participating neuron death in Figure 10c may be explained by early post-synaptic firing, which keeps the network from depressing too heavily. When neuron death occurs, the total average activity exciting the output neuron significantly decreases, resulting in an interval where no post-synaptic spike occurs and excess pre-synaptic spikes depress the network. This may also explain why networks where random neurons were killed failed to recover. The activity of participating neurons was relatively evenly distributed within a pattern window, as was that of non-participating neurons. However, when both sets of neurons die, it is more likely that correlation between spikes in participating and non-participating branches resulted in a significant dip in overall activity which prolongs the time until the next post-synaptic spike. Indeed, Figure 9i shows that when both participating and non-participating branches die together, the post-synaptic neuron sees significantly longer ISIs, if it even manages to fire at all.

Figure 11 shows the analysis of the network when random neurons from the entire population die simultaneously after 30 s of learning. Figure 11a presents the post-synaptic neuron ISI over time, with the left frame looking identical to Figure 8a. As expected, the network quickly loses pattern recognition capability as the % of dead neurons increases, and even struggles in the 5% death case. Figure 11b presents the number of true positive and false positives recognized by the network. The network stops recognizing the pattern and the post-synaptic neuron stops firing almost entirely at 10% afferent death. Interestingly, set 3 in the 5% death case in Figure 11b also shows significant impact, where the other cases maintain a reasonable level of output neuron firing. Figure 11c shows the time of death for the randomly selected afferents. Finally, Figure 11d shows the distribution of dead participating and non-participating afferents in each of the five sets in each case.

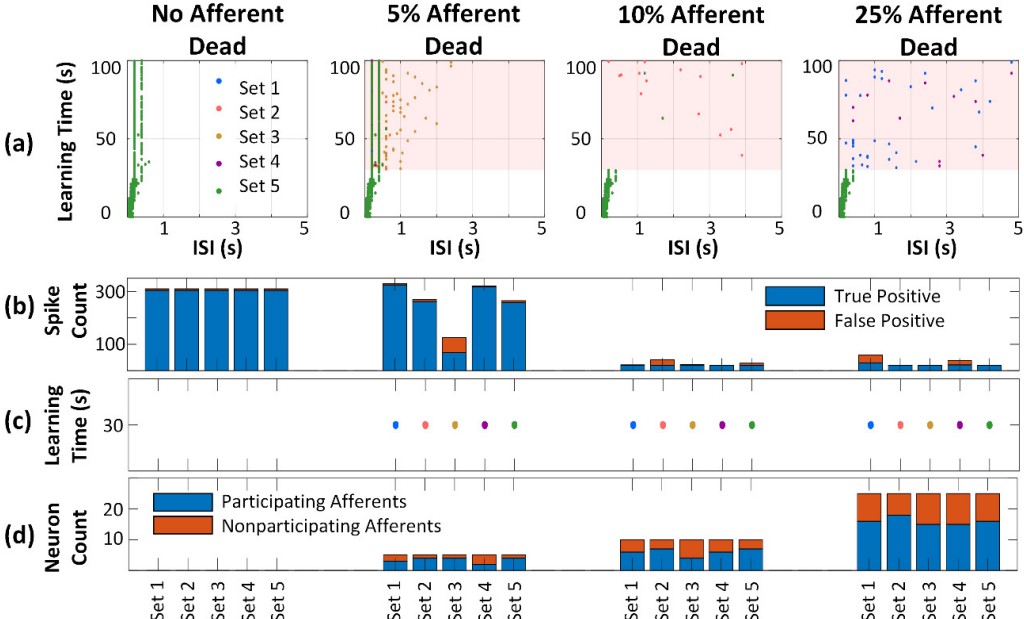

**Figure 11.** (**a**) Post-synaptic neuron ISI over time when all afferent death occurs simultaneously at 30 s. As the % of dead neurons increases, the network loses the pattern recognition capabilities. (**b**) The number of true positive and false positives recognized by the network; the network stops recognizing the pattern and post-synaptic neuron stops firing as neuron death increases. (**c**) All afferents are dead instantaneously at 30 s. (**d**) The distribution of dead participating and non-participating afferents in each of the five sets in each case.

Figure 12 shows a similar analysis of the network as Figure 11, except the pre-synaptic neuron death time occurs randomly (as opposed to simultaneously) starting at 30 s and lasting until 60 s. Figure 12a presents the post-synaptic neuron ISI over time and similarly shows loss in the pattern recognition capabilities of the network as the fraction of failed

neurons increases. Note that in the plotting of these data, some points are obscured by others because they are layered in order of set number. Thus, for the 25% Afferents Dead case in Figure 12a, set 5 actually does stop firing, but sets 1–3 do not (they are mostly obscured by the purple points of Set 4). However, the number of true positives and false positives recognized by the network in Figure 12b shows that death of certain sets of neurons have more detrimental consequences than others. Specifically, sets 2 and 5 in the 15% death case are very nearly unaffected and exhibit a high true positive rate and low false positives. These two sets for the 15% death case do not have different proportions of death between the participating and non-participating populations. Figure 12c,d shows the random neuron death times and the distribution of dead participating and non-participating afferents in each of the five sets in each case, respectively.

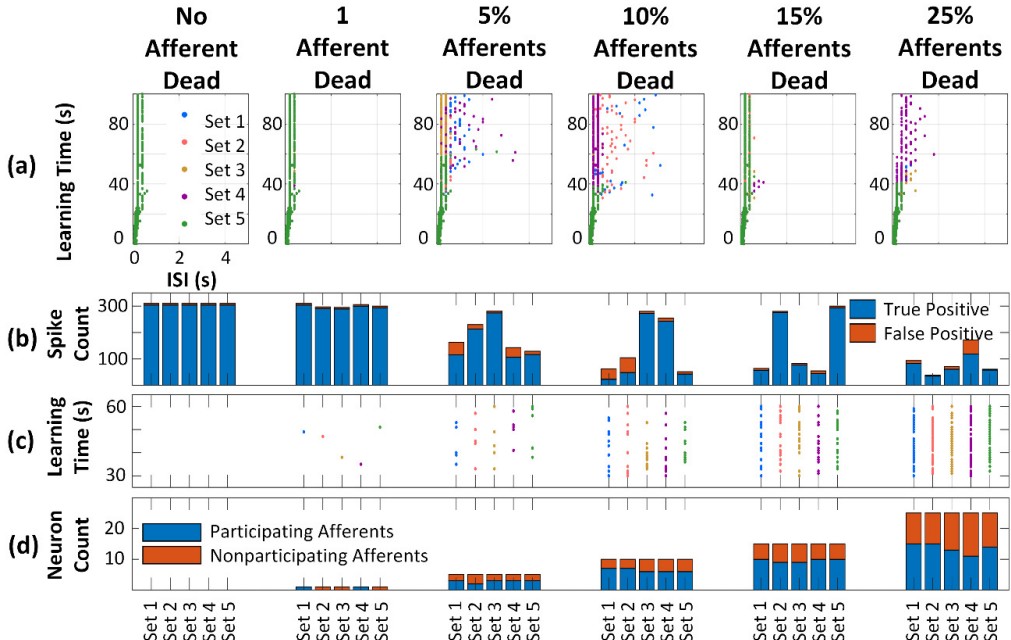

**Figure 12.** (**a**) Post-synaptic neuron ISI over time. As the percentage of dead neurons increases the network loses the pattern recognition capabilities. (**b**) The number of true positive and false positives recognized by the network. In general, the network stops recognizing the pattern and the post-synaptic neuron eventually stops firing as the percentage of failed neurons increases. (**c**) Time of failure of all afferents occurring randomly between 30 s and 60 s into the simulation. (**d**) The distribution of dead participating and non-participating afferents in each of the five sets in each case.

Unlike in Figure 11, where instantaneous afferent death completely disabled the network, in Figure 12, a few sets were able to recover, even in the case of high percentage of neuron failure. Figure 12b demonstrates that at 10% neuron death, the network responded well for set 3 and set 4. Even at 25% neuron death, sets 2 and 4 were not significantly affected. On the other hand, in Figure 11b, the network did not perform well in case of any of the simulated sets with failure percentages of 10% or more. Overall, the network performed better when the afferents failed gradually as compared to instantaneous death. Although the number of simulations carried out here is not large enough to draw conclusions about the fraction of cases in which the network restores itself (or the extent of that restoration), that will be explored in more extensive future studies. Complementary studies will also be undertaken to determine the exact cause for the dramatic differences in pattern recognition outcomes between sets where the same percentages of neurons failed.

## 4. Conclusions

This paper discusses the pattern learning ability of a memristor-based electronic spiking neural network as the afferents in the network fail/die due to external factors such

as radiation exposure, other standard mechanisms such as device or circuit degradation, or other unforeseen events. The general network structure is that of a feed-forward perception with 100 pre-synaptic neurons all connected to a post-synaptic LIF neuron via memristive synapses. The STDP learning rule is implemented using biphasic pulses generated by neurons. The simulations were designed to observe the effect on the learning ability of the network for three cases when selectively only participating neurons are affected, non-participating neurons are disabled, or random/non-selective neuron death occurs after the network is trained, which takes approximately 30 s. In the case of simultaneous neuron failure, network learning ability is least affected when only non-participating afferents are disabled. On the other hand, when random/non-selective neuron death occurs in the network, pattern recognition ability degrades rapidly as 10% (10) of the total afferents are disabled and the network becomes unstable at 5% (5) neuron death. The study also simulates the case when neuron death occurs randomly (instead of simultaneously at 30 s) between 30 s and 60 s of learning. The comparison shows that the network's learning ability is not as seriously deteriorated in the case of gradual neuron death. In some of these cases, the network was still able to recognize the pattern remarkably well, even at 25% neuron failure.

The results of this study show that afferents that do not participate in the pattern still contribute to improving the learning ability of the network, even when partial learning is completed. This emphasizes the importance of non-participating neurons during the learning process. In addition, instantaneous neuron death will degrade the network's pattern recognition capability more than gradual neuron death. However, all cases show that the networks do have some capability to recover and relearn the patterns when undergoing continuous training.

**Author Contributions:** Conceptualization, S.G.; Formal analysis, S.G.; Investigation, S.G.; Project administration, K.D.C.; Software, R.C.I.; Writing—review and editing, R.C.I. and B.R.E. All authors have read and agreed to the published version of the manuscript.

**Funding:** This work was supported by the Defense Threat Reduction Agency (DTRA) grant HDTRA1-17-1-0036.

**Conflicts of Interest:** The authors declare no conflict of interest.

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
