# Peer review of "Neuron Circuit Failure and Pattern Learning in Electronic Spiking Neural Networks"

_electronics, doi:10.3390/electronics11091392_

Round 1

Reviewer 1 Report

High quality paper, with nothing to point.

Author Response

Thank you, we appreciate the positive feedback!

Reviewer 2 Report

Premise: In this work, the authors explore the effects of neuronal death on the accuracy of a spiking neural network. Specifically, they train an SNN to recognize the letter ‘B’, and after 30 seconds, disable various combinations of neurons to assess the impact on the network. They test both sudden neuronal death, as well as gradual death over a period of 30 seconds, for percentages of neurons between a single neuron to up to 25% of participating or non-participating neurons. Their results demonstrate that sudden death has more impact on the ability of the network to recover from the impaired state.

Positive Aspects:

This work is succinct in answering the question it set out to answer. It is well laid out and clearly understood, and does seem to fill a gap in the literature that is not directly addressed in other works. The experiments, if simple, are varied and illustrate well the objectives of the authors.

To be addressed:

The most important aspect to be improved is the missing Section reference found throughout the work. Obviously, this must be corrected before publication.

The authors claim that neuronal death has not been explored in previous literature. This is not entirely true, works [1] and [2] mention that they are trying to address sudden loss of neurons, and [3] states neuronal death as a key learning feature of biological neural networks. Note especially, Fig. 4 of work [2], which states that up to 70% of neurons are eliminated with minimal loss to network accuracy. How do these works integrate with yours?

In relation to above, while it is stated that further analysis will be done in future works, I believe that some analysis of the results is necessary even here, as some results are surprising. Especially, why does eliminating non-participating neurons have an effect on the network if their synaptic weight is in theory repressed? How does the network recover from a loss of neurons? It would be nice to see also the impact of neuronal loss on 2-class classification, aka between ‘A’ and ‘B’.

On page 11, you say that sets 2 and 5 are not significantly affected at 25% neuronal death, but Figure 12-a shows that they stop firing after 40 seconds.

[1] Supervised training of spiking neural networks for robust deployment on mixed-signal neuromorphic processors

[2] Learning arbitrary dynamics in efficient, balanced spiking networks using local plasticity rules

[3] EMERGENCE OF PREFERRED FIRING SEQUENCES IN LARGE SPIKING NEURAL NETWORKS DURING SIMULATED NEURONAL DEVELOPMENT

Author Response

Premise: In this work, the authors explore the effects of neuronal death on the accuracy of a spiking neural network. Specifically, they train an SNN to recognize the letter ‘B’, and after 30 seconds, disable various combinations of neurons to assess the impact on the network. They test both sudden neuronal death, as well as gradual death over a period of 30 seconds, for percentages of neurons between a single neuron to up to 25% of participating or non-participating neurons. Their results demonstrate that sudden death has more impact on the ability of the network to recover from the impaired state.

Positive Aspects:

This work is succinct in answering the question it set out to answer. It is well laid out and clearly understood, and does seem to fill a gap in the literature that is not directly addressed in other works. The experiments, if simple, are varied and illustrate well the objectives of the authors.

To be addressed:

The most important aspect to be improved is the missing Section reference found throughout the work. Obviously, this must be corrected before publication.

Thank you for catching this error that apparently occurred in the pdf conversion process. Given the short length of the paper we elected to remove fields for section references and converted to plain text.

The authors claim that neuronal death has not been explored in previous literature. This is not entirely true, works [1] and [2] mention that they are trying to address sudden loss of neurons, and [3] states neuronal death as a key learning feature of biological neural networks. Note especially, Fig. 4 of work [2], which states that up to 70% of neurons are eliminated with minimal loss to network accuracy. How do these works integrate with yours?

These works provide an interesting addition to the background for our work, and we have referenced [1] and [2] in line 45. In [1] and [2], the networks used contain recurrent connections which the authors claim improve the robustness of their networks to sudden neuron failure. We have included a statement in the introduction referencing these works and clarifying our work as an addition to the literature exploring the effects of neuron death on feed-forward networks.

In [3], it seems the authors are employing a designed neuron death mechanism as a tool for pruning connections during an “early learning” phase of training. This interesting tactic seems to be useful for instantiating the initial connectome in a network, but we feel this method is a bit beyond the scope of the current work.

In relation to above, while it is stated that further analysis will be done in future works, I believe that some analysis of the results is necessary even here, as some results are surprising. Especially, why does eliminating non-participating neurons have an effect on the network if their synaptic weight is in theory repressed? How does the network recover from a loss of neurons? It would be nice to see also the impact of neuronal loss on 2-class classification, aka between ‘A’ and ‘B’.

We very much appreciate the feedback. These are all questions we are very interested in answering as well. The question of the effect of non-participating neurons was also a surprise to us, and we have some suspicions as to what might be going on. We think this may have to do with the distribution of activity on the participating/non-participating branches within a pattern window. Because of the exponential build up on the membrane of the LIF neurons a significant decrease in overall activity at one time may cause the trajectory of the membrane to dip significantly and take a long time to recover to threshold. During this recovery time the network sees a significant amount of pre-synaptic activity which heavily depresses the remaining connections, and if a post-synaptic pulse does not occur within a short enough time window the network will fail to recover. This situation may arise more when random neurons are killed if some non-participating neurons contribute significant activity at the same time as some participating neurons who die. Though the non-participating branches are depressed they still contribute to some baseline level of activity. And it is more likely to have such clusters of spikes dying across branches when both participating and non-participating branches are killed since the encoding for this network necessarily evenly distributes activity between parallel participating branches as well as between parallel non-participating branches, but not necessarily between participating and non-participating branches. This may also explain how the network recovers in some cases but fails to in others. When neurons begin to die off there is a significant decrease in overall membrane potential at the post-synaptic neuron. But if the post-synaptic neuron is able to fire within enough time, the network may be able to bounce back.

We have added the following paragraph starting at line 331 in the manuscript to address this question:

The recovery seen in the average synaptic weight in the network with 25% neuron death in Figure 10 (c) may be explained by early post-synaptic firing which keeps the network from too heavily depressing. When neuron death occurs, the total average activity on the membrane significantly decreases, resulting in a short interval where no post-synaptic spike occurs and excess pre-synaptic spikes begin to depress the network. This may also explain why networks where random neurons were killed failed to recover. The activity of participating neurons was relatively evenly distributed within a pattern window, as was non-participating neurons. However, a subset of each neurons is chosen it is more likely that correlation between spikes in participating and non-participating branches resulted in a significant dip in overall activity which prolonged the time until the next post-synaptic spike. Indeed Figure 9 (i) shows that when both participating and non-participating branches die together the post-synaptic neuron sees significantly longer ISIs, if it even manages to fire at all.

On page 11, you say that sets 2 and 5 are not significantly affected at 25% neuronal death, but Figure 12-a shows that they stop firing after 40 seconds.

That is a great point. Actually, sets 2 and 4 were not significantly affected, but set 5 did stop firing. We have added these details to the manuscript at line 367: “Note that in the plotting of this data, some points are obscured by others because they are layered in order of set number. Thus, for the 25% Afferents Dead case in Figure 12 (a), Set 5 actually does stop firing, but sets 1-3 do not (they are mostly obscured by the purple points of Set 4).”

[1] Supervised training of spiking neural networks for robust deployment on mixed-signal neuromorphic processors

[2] Learning arbitrary dynamics in efficient, balanced spiking networks using local plasticity rules

[3] EMERGENCE OF PREFERRED FIRING SEQUENCES IN LARGE SPIKING NEURAL NETWORKS DURING SIMULATED NEURONAL DEVELOPMENT

Reviewer 3 Report

lines 78,83,86,101,159: invalid link
line 111: Does "s" mean seconds? The first use of this symbol requires an extended description, as the reader has not yet been introduced to the subject of this research.

The problem of learning this network was not discussed. As a result, there is no description of the principle of automatic tuning after the loss of neurons. In order for the system to work up-to-date, there must be such a description with a cause-and-effect block diagram. 

Author Response

lines 78,83,86,101,159: invalid link

Thank you for catching this error that apparently occurred in the conversion to pdf. Given the short length of the paper we elected to remove the section references as fields and converted them to plain text.

line 111: Does "s" mean seconds? The first use of this symbol requires an extended description, as the reader has not yet been introduced to the subject of this research.

We appreciate the feedback. Correct, s was seconds and this has been corrected in the figure caption on line 111.

The problem of learning this network was not discussed. As a result, there is no description of the principle of automatic tuning after the loss of neurons. In order for the system to work up-to-date, there must be such a description with a cause-and-effect block diagram.

In this work, synapses in the system are actually constantly learning using the pair-based STDP learning rule. We have clarified this by adding a statement at line 103. The STDP learning rule can be found in references [18], [19], and [20], the last of which we added because it is our latest open-access publication. We would be happy to add an STDP learning figure to the manuscript similar to figure 5b in reference [20] if the reviewer feels it is necessary.

[20]       S. Gandharava Dahl, R. C. Ivans, and K. D. Cantley, “Effects of memristive synapse radiation interactions on learning in spiking neural networks,” SN Appl. Sci., vol. 3, no. 5, p. 555, May 2021, doi: 10.1007/s42452-021-04553-0.

Round 2

Reviewer 2 Report

The changes are sufficient to answer my questions, all looks good. The only change to make is there is a typo in the new paragraph in the sentence, "However, a subset of each neurons is chosen it is more likely that correlation between spikes in participating and non-participating branches resulted in a significant dip in overall activity which prolonged the time until the next post synaptic spike."

Author Response

We thank the reviewer very much for catching that mistake. We actually went ahead and revised the entire paragraph to make it more clear, as follows:

The recovery seen in the average synaptic weight in the network with 25% non-participating neuron death in Figure 10 (c) may be explained by early post-synaptic firing which keeps the network from too heavily depressing. When neuron death occurs, the total average activity exciting the output neuron significantly decreases, resulting in an interval where no post-synaptic spike occurs and excess pre-synaptic spikes depress the network. This may also explain why networks where random neurons were killed failed to recover. The activity of participating neurons was relatively evenly distributed within a pattern window, as was that of non-participating neurons. However, when both sets of  neurons die, it is more likely that correlation between spikes in participating and non-participating branches resulted in a significant dip in overall activity which prolongs the time until the next post-synaptic spike. Indeed, Figure 9 (i) shows that when both participating and non-participating branches die together, the post-synaptic neuron sees significantly longer ISIs if it even manages to fire at all.